# Design, Synthesis, Electronic Properties, and X-ray Structural Characterization of Various Modified Electron-Rich Calixarene Derivatives and Their Conversion to Stable Cation Radical Salts

**DOI:** 10.3390/molecules27185994

**Published:** 2022-09-14

**Authors:** Rajendra Rathore, Sergey V. Lindeman, Sameh H. Abdelwahed

**Affiliations:** 1Department of Chemistry, Marquette University, Milwaukee, WI 53233, USA; 2Department of Chemistry, Prairie View A&M University, Prairie View, TX 77446, USA

**Keywords:** calixarene, electronic properties, optical properties, cation radical, near-IR region, cyclic voltammetry, ultraviolet-visible-near-infrared (UV-vis-NIR) spectroscopy

## Abstract

We have designed and synthesized electron-rich calixarene derivatives, which undergo reversible electrochemical oxidation in a well-accessible potential range that allows the ready preparation and isolation of the corresponding cation radicals. Preparation of mono- or tetra-radical cation can be achieved by using stable aromatic cation-radical salts such as MA^+•^, MB^+•^, and NAP^+•^ as selective organic oxidants. The cation radicals of calixarenes are stable indefinitely at ambient temperatures and can be readily characterized by UV-vis-NIR spectroscopy. These cation radicals bind a single molecule of nitric oxide within its cavity with remarkable efficiency.

## 1. Introduction

Calix[4]arene derivatives play a fundamental role in the field of supramolecular chemistry [1,2,3], as they are useful in the synthesis of receptors for cations [4,5,6], anions [7,8,9,10], and neutral [11,12,13] molecules. Calix[4]arene is readily synthesized by a one-pot reaction using 4-*tert*-butylphenol and paraformaldehyde in the presence of sodium hydroxide as a catalyst [14]. Due to the strong hydrogen bonding among the hydroxyl groups on the lower rim, unmodified calix[4]arene implement a cone conformation. However, the calixarene tetra-methyl ether derivative is a conformationally mobile molecule with four rapidly inter-converting conformers at room temperature [15,16], whereas the introduction of bulky substitutions into OH groups [17,18,19,20,21,22,23] or by cross-linking two phenyl units [24,25,26,27,28,29,30,31] freezes the rotation of the phenyl units. Therefore, one can synthesize and isolate four calix[4]arene conformers (cone, partial cone, 1,2-alternate, and 1,3-alternate) [32,33]. Rathore et al. [34] have provided the most convincing evidence for the binding of cationic guests within the interior of π-basic cavities of calixarenes by an *X*-ray crystallographic study of the various cationic [calixarene/NO^+^] complexes. Further extension [34] of those results have led to the design and synthesis of a modified calixarene derivative, in which the incorporation of 2,5-dimethoxytolyl as an electron donor on the calixarene allows for the isolation of a stable cation radical salt that binds nitric oxide (NO) deep within its cavity with remarkable efficiency. It is expected that the incorporation of the same electro-active aryl groups, in different calixarene ether conformers, will allow the hole to be delocalized over the aryl groups of the calixarene core, which allows the opportunity to isolate cation radicals for the corresponding calixarene (i.e., Figure 1).

Herein, we now explore our work to prepare various conformers of calix[4]arene ether derivatives (cone and 1,3-alternate, Figure 1) that allow the preparation of a stable cation radical through a chemical or equivalent electrochemical oxidation.

## 2. Results and Discussion

**Synthesis of Modified Arylcalix[4]arene Ethers**. The monoarylcalixarene ether derivatives **A** and **B** both can be synthesized in a six-step reaction sequence starting with the commercially available tert-butylcalix[4]arene (**1**). The principle reaction of the sequence is the coupling of the 2,5-dimethoxytolyl Grignard reagent (derived from 4-bromo-2,5-dimethoxytoluene [35] and Mg in anhydrous tetrahydrofuran) with **6** (or **8**) in the presence of a palladium catalyst affording the corresponding Calixarene **A** (or **B**) in excellent yield (Figure 2).

The (1,3-alternate) diarylcalix[4]arene ether (**C**) was obtained as summarized in Figure 3. Moreover, the synthesis of various conformers of tetraarylcalix[4]arene ethers donors (cone and 1,3-alternate) can be achieved, thus the parent *t*-butylcalix[4]arene (**1**) can be converted to various rigid conformers (**D** & **E**) of calixarene tetrapropyl ether by carefully choosing the base, alkyl halides or tosylate, and solvent in each case, according to the known literature procedures, the details of which are summarized in Figure 4.

The structures of these calixarene donors **A**–**E** were readily established by ^1^H/^13^C NMR spectroscopy and were confirmed by X-ray crystallography, which demonstrated that the shape or the size of the cavity did not change by replacement of *tert*-butyl group(s) with 2,5-dimethoxytolyl (Ar) group(s) in any significant way.

**Electrochemical Studies of the Modified Arylcalix[4]arene Ethers**. The donor strength and the initial indication of the stability of various calixarene cation radicals were evaluated by cyclic voltammetry and were compared with its model compound (**M,**Figure 2). The model compound **M** and various calixarene ethers were oxidized electrochemically at a platinum electrode as 2 × 10^−3^ M solutions in anhydrous dichloromethane containing 0.1 M tetra-*n*-butylammonium hexafluorophosphate (TBAH) as the supporting electrolyte. A reversible cyclic voltammogram (CV) of **M** was obtained at a scan rate of 200 mV s^−1^, and it demonstrated anodic/cathodic peak current ratios of *i*_a_/*i*_c_ = 1.0 (theoretical) at 22 °C (see Figure 2). A quantitative evaluation of the CV peak currents with added (equimolar) ferrocene (as an internal standard, *E*_ox_ = 0.45 V vs. SCE) revealed that the reversible cyclic voltammogram of the compound **M** with only one electro-active group, corresponds to the production of a monocation radical of **M**, i.e., Equation (1).
(1)$$M\overset{E_{1/2}}{⇄}M^{+•}\;+\;e$$

The mono-arylated calixarene donors **A** and **B** demonstrated a single reversible wave, whereas the diarylcalixarene **C** demonstrated two well-identifiable waves in their cyclic voltammograms (see Figure 2). Because of the presence of multiple aryl groups in tetrarylcalixarenes **D**, **E**, their cyclic voltammograms (CV) demonstrated overlapping (distorted) multiple reversible waves and are shown in Figure 2. Unfortunately, a quantitative evaluation of the number of electrons involved in each CV peak of various tetraarylcalixarenes with added ferrocene (internal standard) could not be determined due to the extensive overlap (as well as distortion) of the waves.

The first oxidation potentials (*E*_1/2_) of various arylated-calixarene derivatives **A**–**E**, obtained after calibration of the CV peak positions with ferrocene, were slightly lower (~30–50 mV), as compared to the model donor (**M**, *E*_1/2_ = 1.13 V vs. SCE), and are compiled in Table 1. Interestingly, the presence of multiple waves in the CV’s of **D** and **E** (Figure 2) suggests that the oxidation of one of the electro-active aryl groups in **D** and **E** affects the removal of electrons from other aryl groups. Such an observation indicates that the aryl groups are electronically coupled to each other via the calixarene framework. Moreover, the magnitude of such a coupling can be quantified from electrochemical data, if one takes the observed splitting of the oxidation waves as a qualitative indicator of the resonance energy of this electronic coupling [36], i.e., Figure 1.

**Generation and Spectral Analysis of the Cation-Radicals of Various Arylated-Calixarene Donor.** All the calixarene electron donors listed in Table 1 undergo reversible electrochemical oxidation in the range: E_o_°^x^ = 0.98–1.4 V vs. SCE. The electrochemical reversibility in such a well-accessible potential range allows preparation of the corresponding cation-radicals in dichloromethane solution by either chemical or electrochemical oxidation that can be readily carried out by electron exchange with stable aromatic cation-radical salts [37] **MA^+^**^•^, **MB^+^**^•^, and **NAP^+^**^•^, as selective organic oxidants with reversible reduction potentials that differ by only 230 mV (Figure 3).

Similarly, a mixture of 2,3,5,6-tetrachloro-*p*-benzoquinone (chloranil)/methane-sulfonic acid [38,39,40] or 2,3-dichloro-5,6-dicyano-*p*-benzoquinone (DDQ)/trifluoroacetic acid [40] can also serve as an effective oxidant for the preparation of dichloromethane solutions of the cation-radicals of the various calixarene donors (**A**–**E**). Moreover, the isolation of cation-radical salts from a number of calixarene donors can be effected with triethyloxonium hexachloroantimonate [4] as the 1-electron oxidant (see Appendix A). Alternatively, the stable cation-radical solutions from hard-to-oxidize calixarene donors (**W**, **X**) can be obtained by anodic oxidation in anhydrous dichloromethane at low temperatures (−30 °C) containing tetra-*n*-butylammonium hexafluorophosphate as the supporting electrolyte [41].

To confirm the multiple electron-transfer processes in the tetrarylcalixarene donors, as inferred from the electrochemical data above, their oxidation to mono-cation radicals as well as multiple-charged cation radicals were carried out using various aromatic oxidants in Figure 3. Accordingly, the mono-cation radicals of **A**–**E** could be prepared by using the stable aromatic cation radical salts **MA^+^**^•^ SbCl_6_^−^ and/or **MB^+^**^•^ SbCl_6_^−^. For example, Figure 4a shows the spectral changes attendant upon the reduction of 1.8 × 10^−4^ M **MA^+^**^•^ [λ_max_ (log ε) = 516 nm (3.86)] by an incremental addition of 8.2 × 10^−3^ M monoarylcalixarene **B** to its cation radical **B**^+•^ in dichloromethane at 22 °C. The presence of well-defined isosbestic points at λ_max_ = 458 and 544 nm in Figure 4a established the uncluttered character of the electron transfer. Furthermore, a plot of the depletion of **MA^+^**^•^ (i.e., decrease in the absorbance at 516 nm) and formation of **B**^+•^ (i.e., increase in the absorbance at 1460 nm) against the increments of added **B** (inset, Figure 4a), established that **MA^+^**^•^ was completely consumed after the addition of one equivalent of **B**; and the resulting absorption spectrum of **B**^+•^ [λ_max_ (log ε) = 370 (4.00), 620, 1420 nm] remained unchanged upon the further addition of neutral **B** (i.e., Equation (2).
**MA**^+•^ + **B = B**^+•^ + **MA**(2)

The green-colored **B**^+•^ obtained in Equation (2) is highly persistent at room temperature and did not demonstrate any decomposition during a 12 h period at 22 °C. Similarly, the conversion to the corresponding mono-cation radical of tetraarylcalixarene **E** could be carried out by oxidation with **MA^+•^** and is shown in Figure 4b.

Moreover, redox titrations with the monoarylcalixarenes (**A** and **B**), the diarylcalixarene (**C**) and tetraarylcalixarenes (**D** and **E**) established that they can be oxidized selectively to the stable solutions of mono-cation radicals using **MA**^+•^ and/or **MB^+•^** as 1-electron oxidants (Appendix A). The data (absorption maxima and the molar extinction coefficients) obtained from UV-vis-NIR spectral analysis are compiled in Table 2, and the absorption spectra of various calixarene ether mono-cation radicals together with the spectrum of the cation radical of model donor **M** are compared in Figure 5a,b. However, oxidation of tetraarylcalixarenes (**D** and **E**) with stronger oxidant **NAP**^+•^ SbCl_6_^−^ (*E*_red_ = 1.34 V vs. SCE) produced solutions of stable tetracation-radical salts that were stable for several hours at ambient temperatures. For example, when a dark blue solution of **NAP**^+•^ (λ_max_ = 672, 616, 503, and 396 nm; ε_672_ = 9300 M^−1^ cm^−1^) was mixed with 1/4 equiv. of neutral **E**, a dramatic color change to bright green [λ_max_ (log ε) = 400 (4.47), 600, and 1240 nm] occurred immediately, i.e., Equation (3).
**NAP^+^**^•^ + 1/4 **E**
**=** 1/4 **E**^4**+**•^ + **NAP**(3)

It is noteworthy that the absorption spectrum of tetracation radical **E**^4+•^ obtained above was identical to that obtained by an oxidation of **E** using the DDQ/CF_3_COOH method. The UV-vis spectral analysis established the simultaneous oxidation of **E** and reduction of **NAP**^+•^ in quantitative yields, and the uncluttered character of the electron transfer was established by the presence of well-defined isosbestic points at λ_max_ = 414, 480, and 710 nm when a solution of **NAP**^+•^ was treated with incremental amounts of **E** (Figure 6). Furthermore, a plot of the depletion of **NAP^+^**^•^ (i.e., decrease in the absorbance at 672 nm) and formation of **E**^4+•^ (i.e., increase in the absorbance at 1240 nm) against the increments of added **E** in Figure 6 (inset), established that **NAP^+•^** was completely consumed after the addition of 1/4 equiv. (i.e., Equation (3)). Such a spectral titration clearly established the formation of a tetra cation-radical salt from **E** (Figure 6). The UV-vis absorption data for various tetra-cation radicals (from **D** and **E**), obtained by a careful titration method using **NAP^+•^**, are summarized in Table 2, and the spectras are compared in Figure 5c.

The oxidation states of the mono-cation radicals **A**^+•^- **E**^+•^ and multiply-charged cation radicals **D**^4+•^and **E**^4+•^ were further confirmed by the oxidation of a pale-yellow solution of octamethylbiphenylene [42] (**OMB**, *E*_red_ = 0.8 V vs. SCE) with either 1 equiv. of [**E**^+•^] or 1/4 equiv. of [**E**^4+•^] in dichloromethane to a dark blue solution of **OMB**^+•^ (λ_max_ = 602 nm, ε_1105_ = 19,000 M^−1^cm^−1^) in quantitative yield as estimated by the spectral analysis in Figure 7 (e.g., Equations (4) and (5)).
**OMB** + **E**^+•^ → **OMB**^+•^ + **E**(4)
**OMB** + 1/4 **E**^4**+**•^
**→ OMB^+^**^•^ + 1/4 **E**(5)

The mono-cation radicals of tetra-arylcalixarenes (**D** and **E**) demonstrated the characteristic (UV-vis-NIR) absorption spectra (in Figure 5c) with three absorption bands I, II, and III. These spectra strongly resemble the absorption spectrum of the model **M**^+•^ with a slight red shift of band III in various calixarene (**D**, **E**) cation radicals (Figure 5b). Interestingly, the cation radicals derived from mono- and di-arylcalixarenes (**A**–**C**) showed bands I and II together with a structured band III as well as an hyperchromic increase in the (relative) intensity of band III as compared to the model cation radical **M**^+•^. Such an observation is indicative of an increased electronic coupling of the cationic aryl groups with the calixarene core (Figure 5a). Similarly, the various tetra-cation radicals in Figure 5c demonstrated a (broad) absorption band (band III, in near-IR region) in addition to the absorption bands I and II in UV-vis region. It is important to emphasize that the observation of the NIR absorption band III in calixarene cation radicals is highly indicative of the electronic coupling amongst various aryl groups (as well as with the electro-active calixarene core) via both charge delocalization and intramolecular electron hopping. The mono-cation radical of tetra-arylcalixarene **E** was isolated and confirmed by X-ray.

Binding of Nitric Oxide (NO) to Various Modified Calixarene Donors. The ready availability of these calixarene cation-radical salts, with electron-deficient cavities owing to the electronic coupling between charge-bearing aryl moieties and calixarene cores, allows us to directly examine the spectral and structural changes attendant upon association of nitric oxide; for example, when a dichloromethane solution of the isolated monoarylcalixarene ether cation radical [B^+•^ SbCl_6_^−^] (λ_max_ = 1460, 630, and 386 nm, ε_1460_ = 5600 M^−1^ cm^−1^) was exposed to gaseous nitric oxide (NO), the green color was immediately replaced by a dark blue coloration, and the UV-vis spectral analysis of the resulting solution demonstrated a characteristic absorption spectrum with a broad absorption band at λ_max_ = 590 nm (ε_590_ = 8860 M^−1^ cm^−1^) [34] (Figure 8). Note that the observed (UV-vis) spectrum of [B/NO]^+^ association is characteristically similar to that observed for [X/NO]^+^ complex of *t*-butylcalixarene tetrapropyl ether (λ_max_ = 569 nm, ε_569_ = 5600 M^−1^ cm^−1^) [2].

In another experiment [34], a solution of **B** was added to a solution of NO^+^ SbCl_6_^−^ in dichloromethane in a Schlenk flask under an argon atmosphere. The solution immediately takes on a dark-blue coloration and the solution was stable at room temperature for several days. The (UV-vis) spectral analysis of the blue-colored solution shows an absorption spectrum (λ_max_ = 590 nm) identical to that obtained above by exposure of B^+•^ to gaseous NO. Such observation confirms that NO is encapsulated inside the cationic hole of calixarene cavity. (See the supporting information for the X-ray structure [34] [E/NO]^+^).

## 3. Materials and Methods

### 3.1. Cyclic Voltammetry

Cyclic Voltammetry (CV) was performed on an Epsilon E2 Electrochemical Analyzer (Bioanalytical Systems). The CV cell was of an air-tight design with high vacuum Teflon valves and Viton O-rings seals to allow an inert atmosphere to be maintained without contamination by grease. The working electrode consisted of an adjustable platinum disk embedded in a glass seal to allow periodic polishing (with a fine emery cloth) without changing the surface area (~1 mm^2^) significantly. The reference SCE electrode (saturated calomel electrode) and its salt bridge were separated from the catholyte by a sintered glass frit. The counter electrode consisted of a platinum gauze that was separated from the working electrode by ~3 mm.

The CV measurements were carried out in a solution of 0.2 M supporting electrolyte (tetra-*n*-butylammonium hexafluorophosphate, TBAH) and 2–5 × 10^−3^ M substrate (or electron donor) in dry dichloromethane under an argon atmosphere. All the cyclic voltammograms were recorded at a sweep rate of 200 mV sec^−1^, unless otherwise specified and were IR compensated. The oxidation potentials (*E_1/2_*) were referenced to SCE, which was calibrated with added (equimolar) ferrocene (*E_1/_*_2_ = 0.450 V vs. SCE). The *E_1/2_* values were calculated by taking the average of anodic and cathodic peak potentials in the reversible cyclic voltammograms.

The donor strength and the initial indication of the stability of various calixarene cation radicals were evaluated by cyclic voltammetry and were compared with its model compound (**M**). The model compound **M** and various calixarene ethers were oxidized electrochemically at a platinum electrode as 2 × 10^−3^ M solutions in anhydrous dichloromethane containing 0.1 M tetra-*n*-butylammonium hexafluorophosphate (TBAH) as the supporting electrolyte. A reversible cyclic voltammogram (CV) of **M** was obtained at a scan rate of 200 mV s^−1^ and it showed anodic/cathodic peak current ratios of *i*_a_/*i*_c_ = 1.0 (theoretical) at 22 °C. A quantitative evaluation of the CV peak currents with added (equimolar) ferrocene (as an internal standard, *E*_ox_ = 0.45 V vs. SCE) revealed the reversible cyclic voltammogram of the compound **M.**

### 3.2. Oxidation of Various Calixarene Donors

#### 3.2.1. Stoichiometric Oxidation of Tetraarylcalixarene Using MA^+•^ as an Oxidant

A stock solution of **MA**^+**•**^ (λmax = 516 nm, ε_516_ = 7300 M^−1^ cm^−1^) was prepared by dissolving a known quantity of [**MA**^+**•**^SbCl_6_^−^] (2.66 mg, 4.4 × 10^−3^ mmol) in anhydrous dichloromethane (20 mL) at 22 °C and under an argon atmosphere. A 3 mL aliquot of the red-orange solution was transferred to a 1 cm quartz cuvette equipped with a Schlenk adapter (under an argon atmosphere). The redox titrations were carried out by adding the increments of the tetraarylcalixarene donor, for example, **E** dissolved in dichloromethane (4.1 × 10^−3^ M) to the above solution of **MA**^+**•**^, and the accompanied color changes were monitored by UV-vis-NIR spectroscopy. Note that a complete disappearance of **MA**^+**•**^ and the formation of a green colored species (λmax = 375, 620, and 1200 nm, ε_375_ = 6025 M^−1^ cm^−1^) was observed upon the addition of one equivalent of tetraarylcalixarene donor **E**. The addition of **E** beyond 1 equivalent did not demonstrate any changes in the spectra. Solutions of the tetraarylcalixarene cation radical **E**^+**•**^ were routinely prepared by mixing a 1:1 solution of **MA**^+**•**^ with **E** in dichloromethane at 22 °C. Thus, a reaction of 0.02 M **MA**^+**•**^ with 0.1 M **E** in dichloromethane at 22 °C immediately afforded a bright green solution and the UV-Vis spectral analysis revealed the formation of **E**^+**•**^ in quantitative yield.

Note that a treatment of above green solution of **E**^+**•**^ with a pinch of zinc dust immediately led to colorless solution from which a neutral tetraarylcalixarene donor **E** was recovered quantitatively by a simple filtration through a pad of silica gel.

#### 3.2.2. Preparation of 1,2,3,4,7,8,9,10-octahydro-1,1,4,4,7,7,10,10-octamethylnaphthacene Cation Radical Salt (NAP^+•^ SbCl_6_^−^)

Neutral (hindered) naphthalene **NAP** was prepared according to a literature procedure [43] and recrystallized from benzene to afford a colorless crystalline solid in 88% yield, mp 319–320 °C (mp 319–320 °C), ^1^H NMR (CDCl_3_) δ 1.36 (s, 24H), 1.73 (s, 8H), 7.68 (s, 4H); ^13^C NMR (CDCl_3_) δ 32.55, 34.47, 35.16, 123.98, 130.25, 143.42. GCMS: *m/z* 348 (M^+^), 348 calcd. for C_28_H_36_.

A 50 mL flask fitted with a Schlenk adapter was charged with nitrosonium hexachloroantimonate (183 mg, 0.50 mmol), and a prechilled (~−10 °C) solution of naphthalene **NAP** (172 mg, 0.50 mmol) in anhydrous dichloromethane (20 mL) was added under an argon atmosphere. The nitric oxide produced was entrained by bubbling argon through the solution, which upon spectrophotometric analysis indicated the quantitative formation of **NAP^+•^** SbCl_6_^−^. The dark blue solution was carefully layered with dry hexanes (30 mL) and placed in a refrigerator (~−10 °C). For 2 days, dark blue crystals of the cation radical salt were deposited. The crystals were filtered under an argon atmosphere and dried in vacuo (302 mg, 0.45 mmol).

#### 3.2.3. Preparative Isolation of Stilbenoid Cation Radical Salts Using Et_3_O^+^ SbCl_6_^−^

A 200 mL flask equipped with a Schlenk adapter was charged with triethyloxonium hexachloroantimonate (65.7 mg, 0.15 mmol), and a solution of the 1 equiv. of calixarene donor (e.g., **B**) in anhydrous dichloromethane (20 mL) was added under an argon atmosphere at 0 °C. The heterogeneous mixture immediately took on a green coloration which intensified with time. The dark mixture was stirred for 1 h to yield a green solution of **B^+^****^∙^**. The dark green solution was cooled to −50 °C in a dry ice-acetone bath, and anhydrous hexanes (50 mL) were added to precipitate the dissolved salt. The dark-colored precipitate was filtered under an argon atmosphere, washed with hexane (3 × 25 mL), and dried in vacuo. The cation radical **B^+.^** SbCl_6_^−^ (*vide infra*) was obtained in an essentially quantitative yield (0.13 mmol).

The purity of the isolated cation radical **B**^+**•**^ SbCl_6_^−^ was determined by iodometric titration as follows. A solution of **B**^+**•**^ SbCl_6_^−^ in dichloromethane was added to a dichloromethane solution containing excess tetra-*n*-butylammonium iodide at 22 °C, under an argon atmosphere to afford a dark brown solution. The mixture was stirred for 5 min and was titrated (with rapid stirring) by a slow addition of a standard aqueous sodium thiosulfate solution (0.005 M) in the presence of a starch solution as an internal indicator. Based on the amount of thiosulfate solution consumed, purity of the cation radical was determined to be >97%. With the same procedure, the crystalline cation radical salts of other calixarene ethers can be isolated.

### 3.3. Chemistry

The instruments used for measuring the melting points, spectral data Mass, ^1^H NMR, and ^13^C NMR are provided in detail in the Appendix A.

**11,17,23-Tri-*p*-*tert*-butyl-25,26,27,28-tetrapropoxy calix[4]arene (5). (Cone Conformer).** The titled compound was synthesized by using 11,17,23-Tri-*p*-*tert*-butyl-25,26,27,28-tetrahydroxy calix[4]arene (**4**), thus, **4** (3.0 g, 5.1 mmol) and *n*-PrI (16 mL, 153 mmol) were dissolved in DMF (60 mL), and the solution was stirred at room temperature overnight in the presence of Ba(OH)_2_.8H_2_O (5.60 g, 17.85 mmol) and BaO (1.19 g, 7.80 mmol). The reaction mixture was diluted with water (300 mL) and extracted with dichloromethane (3 × 100 mL). The organic layer was separated and dried over anhydrous magnesium sulfate. After filtration through a short pad of silica gel, the filtrate was concentrated to dryness. The residue was triturated with methanol (50 mL) to afford **5**. Yield: 3.2 g (83%), mp 181–182 °C (CH_2_Cl_2_/MeOH); ^1^H NMR (CDCl_3_) δ: 0.77 (s, 9H), 0.90 (t, *J* = 7.5 Hz, 6H), 1.08 (m, 6H), 1.35 (s, 18H), 1.89 (sextet, *J* = 7.08 Hz, 4H), 2.04 (sextet, *J* = 7.5 Hz, 4H), 3.08 (s, 2H), 3.12 (s, 2H), 3.65 (t, *J* = 6.8 Hz, 2H), 3.67 (t, *J* = 6.8 Hz, 2H), 3.98 (m, 4H), 4.35 (d, *J* = 3.6 Hz, 2H), 4.45 (d, *J* = 3.6 Hz, 2H), 6.19 (s, 2H), 6.23 (s, 3H), 7.05 (d, *J* = 2.3 Hz, 2H), 7.09 (d, *J* = 2.3 Hz, 2H); ^13^C NMR (CDCl_3_) δ: 10.17, 11.14, 11.19, 23.37, 23.79, 32.88, 31.35, 31.48, 31.48, 32.03, 33.73, 34.34, 76.68, 77.20, 77.33, 12.46, 124.47, 125.46, 125.97, 127.28, 132.15, 133.45, 135.65, 136.15, 143.82, 144.42, 152.89, 155.07, 155.22. MS (ESI) calculated mass for the parent C_52_H_72_O_4_ 761.13, found 762.14 [M + 1]^+^.

**11,17,23-Tri-*p*-*tert*-butyl-25,26,27,28-tetrapropoxycalix[4]arene(1,3-Alternate Conformer) (7).** A mixture of **4** (2.0 g, 3.4 mmol) and Cs_2_CO_3_ (33.2 g, 102 mmol) in DMF (50 mL) was heated at 80 °C for 60 min. Subsequently, *n*-propyl tosylate (21.8 g, 102 mmol) was added and the reaction mixture was heated to 80 °C for 7 h. Upon cooling, the reaction mixture was poured into water (300 mL). After extraction with dichloromethane (3 × 100 mL) the combined organic fractions were washed with 1 N HCl (1 × 50 mL) and brine (3 × 50 mL). The organic layer was dried over magnesium sulfate and evaporated under reduced pressure. To remove the excess *n*-propyl tosylate, a mixture of the resulting residue, potassium iodide (4.0 g), and triethyl amine (4 mL) in acetonitrile (100 mL) was refluxed for 1 h. After removal of the solvent, dichloromethane (300 mL) was added to the residue, and the organic layer was washed with 1 N HCl (2 × 100 mL) and water (2 × 100 mL). The crude solid thus obtained was subjected to flash chromatography using silica gel and a 75:25 mixture of hexanes/ethyl acetate as eluent. Yield: 1.3 g (52%), mp 231–233 °C (CH_2_Cl_2_/MeOH); ^1^H NMR (CDCl_3_), δ: 0.70 (m, 12H), 1.11–1.33 (m, 8H), 1.25 (s, 18H), 1.27 (s, 9H), 3.38 (m, 4H), 3.73 (s, 4H), 3.74 (s, 4H), 6.72 (t, *J* = 7.5 Hz, 1H), 6.94–7.00 (m, 8H); ^13^C NMR (CDCl_3_) δ: 10.48, 10.52, 10.73, 22.92, 23.01, 23.33, 31.85, 31.89, 34.15, 34.16, 38.51, 38.92, 72.40, 72.45, 72.51, 125.15, 126.03, 126.11, 126.28, 129.56, 132.83, 132.88, 133.11, 133.98, 143.34, 143.41, 154.52, 154.75, 154.95. MS (ESI) calculated mass for the parent C_52_H_72_O_4_ 761.13, found 762.14 [M + 1]^+^.

**General Procedure for Mono-Bromination of Calix[4]arene (6 and 8).** To a solution of appropriate calixarene (1 mmol) in methylethyl ketone (35 mL) was added *n*-bromosuccinimide (2.2 mmol) and the yellow solution was stirred at room temperature for 24 h. The reaction mixture was then treated with 10% NaHSO_3_ (30 mL) and worked up by extraction with dichloromethane (3 × 15 mL). The combined organic extracts were dried over anhydrous magnesium sulfate and evaporated under reduced pressure to afford a cream-colored solid. The resulting crude material was purified by flash chromatography using silica gel and a 95:5 mixture of hexanes and ethyl acetate as an eluent to afford pure product.

**5-Bromo-11,17,23-tri-*p*-*tert*-butyl-25,26,27,28-tetrahydroxy calix[4]arene (Cone Conformer) (6)**. Yield: (86%), mp 222–223 °C (CH_2_Cl_2_/MeOH); ^1^H NMR (CDCl_3_) δ: 0.81 (s, 9H), 0.91 (t, *J* = 7.53 Hz, 6H), 1.07 (m, 6H), 1.33 (s, 18H), 1.89 (sextet, *J* = 6.7 Hz, 4H), 2.04 (m, 4H), 3.06 (m, 4H), 3.62 (t, *J* = 7.05 Hz, 2H), 3.69 (t, *J* = 6.9 Hz, 2H), 3.96 (m, 4H), 4.39 (t, *J* = 13.14 Hz, 4H), 6.29 (s, 2H), 6.45 (s, 2H), 7.00 (d, *J* = 2.25 Hz, 2H), 7.11 (d, *J* = 2.25 Hz, 2H); ^13^C NMR (CDCl_3_) δ: 10.27, 11.09, 11.26, 23.43, 23.76, 23.97, 31.27, 31.54, 32.08, 33.92, 34.47, 114.69, 124.71, 125.22, 126.39, 130.37, 132.16, 134.72, 135.97, 136.24, 144.44, 144.97, 152.80, 154.45, 154.98. MS (ESI) calculated mass for the parent C_52_H_71_BrO_4_ 840.02, found 841.14 [M + 1]^+^.

**5-Bromo-11,17,23-tri-*p*-*tert*-butyl-25,26,27,28-tetrahydroxy calix[4]arene (1,3-alternate Conformer) (8).** Yield: (82%), mp 261–262 °C (CH_2_Cl_2_/MeOH); ^1^H NMR (CDCl_3_) δ: 0.64 (t, *J* = 7.59 Hz, 3H), 0.76 (t, *J* = 7.41 Hz, 9H), 1.02 (sextet, *J* = 7.83 Hz, 2H), 1.25 (s, 18H), 1.28 (m, 6H), 1.36 (s, 9H), 3.36 (m, 8H), 3.70 (s, 4H), 3.76 (s, 4H), 6.97 (s, 4H), 6.98 (s, 2H), 7.13 (s, 2H); ^13^C NMR (CDCl_3_) δ: 10.36, 10.57, 10.75, 22.58, 22.88, 23.12, 31.85, 31.89, 34.15, 34.19, 38.23, 39.15, 72.02, 72.37, 72.57, 114.26, 126.06, 126.18, 126.26, 131.96, 132.19, 132.74, 133.26, 136.31, 143.64, 154.32, 154.65, 155.42. MS (ESI) calculated mass for the parent C_52_H_71_BrO_4_ 840.02, found 841.04 [M + 1]^+^.

**General Procedure for Di-Bromination of Calix[4]arene.** To a solution of appropriate calixarene (1 mmol) in methylethyl ketone (35 mL) was added *n*-bromosuccinimide (4.4 mmol), and the yellow solution was stirred at room temperature for 24 h, according to the general procedure described above.

**11,23-Dibromo-5,17-di-*tert*-butyl-25,27-dihydroxy- 26,28-dipropoxy calix[4]arene (12).** Yield: (88%); ^1^H NMR (CDCl_3_) δ: 1.07 (s, 18H), 1.29 (t, *J* = 7.5 Hz, 6H), 2.10 (sextet, *J* = 7.8 Hz, 4H), 3.32 (s, 4H), 3.98 (t, *J* = 7.5 Hz, 4H), 4.32 (s, 4H), 6.94 (s, 4H), 7.20 (s, 4H), 8.50 (s, 2H); ^13^C NMR (CDCl_3_) δ: 10.41, 10.79, 22.48, 23.12, 31.55, 34.11, 38.63, 72.22, 72.47, 114.23, 126.13, 126.20, 132.16, 132.49, 134.26, 148.61, 155.41. MS (ESI) calculated mass for the parent C_48_H_62_Br_2_O_4_ 861.86, found 862.88 [M + 1]^+^.

**General Procedure for Tetra-Bromination of Calix[4]arene Derivatives.** To a solution of appropriate calixarene (1 mmol) in methylethyl ketone (35 mL) was added *n*-bromosuccinimide (8.8 mmol), and the yellow solution was stirred at room temperature for 24 h, according to the general procedure described above.

**5,11,17,23-Tetrabromo-25,26,27,28-tetrapropoxycalix[4]arene (Cone Conformer) (16)**; Yield: (74%); mp 280–282 °C (CH_2_Cl_2_/MeOH) (mp 278–280 °C); ^1^H NMR (CDCl_3_) δ: 0.95 (t, *J* = 7.41 Hz, 12H), 1.91 (sextet, *J* = 7.41 Hz, 8H), 3.10 (d, *J* = 13.4 Hz, 4H), 3.85 (t, *J* = 7.41 Hz, 8H), 4.37 (d, *J* = 13.4 Hz, 4H), 6.81 (s, 8H); ^13^C NMR (CDCl_3_) δ: 11.07, 23.82, 31.36, 77.23, 114.97, 130.66, 136.11, 155.02. MS (ESI) calculated mass for the parent C_40_H_44_Br_4_O_4_ 908.39, found 909.41 [M + 1]^+^.

**5,11,17,23-Tetrabromo-25,26,27,28-tetrapropoxy calix[4]arene (1,3-Alternate Conformer) (17)**; Yield: (67%); mp 251–253 °C (CH_2_Cl_2_/MeOH); ^1^H NMR (CDCl_3_) δ: 1.02 (t, *J* = 7.4 Hz, 12H), 1.73 (sextet, *J* = 7.2 Hz, 8H), 3.58 (s, 8H), 3.60 (t, *J* = 7.021 Hz, 8H), 7.16 (s, 8H); ^13^C NMR (CDCl_3_) δ: 11.33, 24.17, 36.62, 73.61, 114.36, 132.29, 134.54, 154.89. MS (ESI) calculated mass for the parent C_40_H_44_Br_4_O_4_ 908.39, found 909.42 [M + 1]^+^.

**General Procedure for the Kumada-Type Coupling for the Preparation of Tetraarylcalixarene (A and B).** A solution of 2,5-dimethoxy-4-methylphenylmagnesium bromide was prepared from 4-bromo-2,5-dimethoxytoluene (0.66 mmol) and excess magnesium turnings (1.32 mmol) in anhydrous tetrahydrofuran (30 mL), under an argon atmosphere, by refluxing for 4 h. The Grignard reagent thus obtained was transferred at room temperature to another Schlenk flask containing appropriate monobromo-calixarene (0.33 mmol) and a catalytic amount of *bis*(triphenylphosphine)palladium dichloride (50 mg). The resulting yellow mixture was refluxed overnight, cooled to room temperature, and quenched with dilute hydrochloric acid (5%, 30 mL). The aqueous layer was extracted with dichloromethane (3 × 50 mL) and the combined organic extracts were dried over anhydrous magnesium sulfate and filtered. Evaporation of the solvent in vacuo afforded a brown residue that was filtered through a short pad of silica gel using hexanes as eluent. The solid product obtained after removal of the solvent was purified by flash chromatography on silica gel using a 90:10 mixture of hexanes and ethyl acetate as eluent to afford corresponding monoarylcalixarene as a colorless crystalline solid.

**5-(2,5-Dimethoxytoluene)-11,17,23-tri-*p*-*tert*-butyl-25,26,27,28-tetrahydroxy calix[4]arene. (Cone Conformer) (A).** Yield: (89%); mp 203–204 °C (CH_2_Cl_2_/MeOH); ^1^H NMR (CDCl_3_) δ: 1.00 (s, 18 H), 1.01 (m, 12H), 1.06 (s, 9H), 1.99 (sextet, *J* = 7.5 Hz, 4H), 2.04 (sextet, *J* = 7.5 Hz, 4H), 2.24 (s, 3H), 3.15 (t, *J* = 12.3 Hz, 4H), 3.71 (s, 3H), 3.76 (s, 3H), 3.79 (t, *J* = 7.5 Hz, 4H), 3.90 (q, *J* = 6.9 Hz, 4H), 4.45 (dd, *J* = 12.6 Hz, 4H), 6.63–6.80 (m, 8H), 7.10 (s, 2H); ^13^C NMR (CDCl_3_) δ 10.49, 10.59, 10.77, 16.55, 23.47, 23.59, 23.66, 31.39, 31.69, 31.76, 34.04, 56.15, 56.57, 56.83, 77.01,77.20, 77.32. 113.23, 115.33, 124.80, 125.01, 125.19, 125.57, 128.97, 129.19, 131.97, 133.26, 133.52, 134.67, 134.85, 144.07, 144.35, 150.28, 151.69, 153.66, 154.09, 155.62. MS (ESI) calculated mass for the parent C_61_H_82_O_6_ 911.31, found 912.34 [M + 1]^+^. Analysis for C_61_H_82_O_6_ (911.32), Calcd.; %C, 80.40; H, 9.07; Found: %C 80.93; H, 8.21.

**5-(2,5-Dimethoxytoluene)-11,17,23-tri-*p*-*tert*-butyl-25,26,27,28-tetrahydroxy calix[4]arene (1,3-Alternate Conformer) (B).** Yield: (77%); mp 256–257 °C (CH_2_Cl_2_/MeOH); ^1^H NMR (CDCl_3_) δ: 0.58 (m, 12H), 1.06 (m, 8H), 1.27 (s, 27H), 2.26 (s, 3H), 3.32 (m, 8H), 3.73 (s, 3H), 3.78 (s, 3H), 3.83 (s, 4H), 3.86 (s, 4H), 6.76 (s, 2H), 6.77 (s, 2H), 6.97 (s, 4H), 7.11 (s, 2H); ^13^C NMR (CDCl_3_) δ 10.37, 10.43, 10.56, 16.58, 22.56, 22.66, 22.85, 31.88, 34.16, 39.25, 39.52, 56.19, 56.55, 71.75, 71.82, 71.91, 113.40, 114.65, 125.49, 125.79, 129.20, 129.85, 131.96, 132.97, 133.08, 133.53, 143.50, 150.22, 151.59, 154.87, 156.25. MS (ESI) calculated mass for the parent C_61_H_82_O_6_ 911.31, found 912.37 [M + 1]^+^. Analysis for C_61_H_82_O_6_ (911.32), Calcd.; %C, 80.40; H, 9.07; Found: %C 81.31; H, 8.82.

**Kumada-Type Coupling for the Preparation of Diarylcalixarene (C).** Prepared by refluxing the dibromo derivative **12** (0.86 g, 1.0 mmol) with 2,5-dimethoxy-4-methylphenylmagnesium bromide (3.0 mmol) in THF (25 mL) in the presence of *bis*(triphenylphosphine)palladium dichloride (50 mg), according to the general procedure described above. The resulting product was purified by flash chromatography using silica gel and a 95:5 mixture of hexanes/ethyl acetate as eluent.

11,23-*Bis*(2,5-dimethoxy-4-methylphenyl)-5,17-di-*tert*-butyl-25,26,27,28-tetrapropoxycalix[4]arene (1,3-Alternate Conformer) (C). Yield: (78%); mp 310–312 °C; ^1^H NMR (CDCl_3_) δ: 0.56 (t, *J* = 7.5 Hz, 6H), 0.72 (t, *J* = 7.44 Hz, 6H), 1.26 (m, 8H), 1.27 (s, 18H), 2.27 (s, 12H), 3.41 (m, 8H), 3.64 (s, 6H), 3.72 (s, 6H), 3.84 (s, 8H), 6.72 (s, 2H), 6.78 (s, 2H), 7.02 (s, 4H), 7.16 (s, 4H); ^13^C NMR (CDCl_3_) δ: 10.18, 10.23, 22.68, 23.01, 31.6, 33.85, 38.71, 55.58, 56.52, 72.13, 72.36, 112.99, 114.87, 125.41, 126.02, 129.13, 130.19, 131.63, 132.79, 133.24, 143.34, 150.11, 151.62, 154.83, 156.18. MS (ESI) calculated mass for the parent C_66_H_84_O_8_ 1005.37, found 1006.39 [M + 1]^+^. Analysis for C_66_H_84_O_8_ (1005.37); Calcd. %C, 78.85; H, 8.42; Found: %C, 77.67; H, 7.83.

**General procedure: Kumada-Type Coupling for the Preparation of Tetraarylcalixarene (D&E).** Prepared by refluxing of appropriate tetraobromo-calixarene (1.1 mmol) with 2,5-dimethoxy-4-methylphenylmagnesium bromide (5.5 mmol) in THF (30 mL) in the presence of *bis*(triphenylphosphine)palladium dichloride (0.1 g), according to the general procedure described above. The resulting product was purified by flash chromatography using silica gel and a 80:20 mixture of hexanes/ethyl acetate as eluent.

**5,11,17,23-*tetrakis*(2,5-Dimethoxy-4-methylphenyl)-25,26,27,28-tetrapropoxycalix[4]arene (Cone Conformer) (D)**. Yield: (54%); mp 229–231 °C (CH_2_Cl_2_/MeOH); ^1^H NMR (CDCl_3_) δ: 1.05 (t, *J* = 7.44 Hz, 12H), 2.12 (sextet, *J* = 7.44 Hz, 8H), 2.15 (s, 12H), 3.22 (d, 4H), 3.31 (s, 12H), 3.43 (s, 12H), 3.95 (t, *J* = 7.5 Hz, 12H), 4.65 (d, *J* = 12.8 Hz, 4H), 6.57 (s, 4H), 6.78 (s, 4H), 7.10 (s, 8H); ^13^C (CDCl_3_) δ: 11.20, 16.92, 24.50, 31.79, 55.73, 56.25, 112.67, 114.42, 124.90, 128.09, 131.83, 133.90, 149.22, 150.98, 155.46. MS (ESI) calculated mass for the parent C_76_H_88_O_12_ 1193.54, found 1193.55 [M + 1]^+^. Analysis for C_76_H_88_O_12_ (1193.5) Calcd. %C, 76.48; H, 7.43; Found %C, 75.63; H, 8.22.

**5,11,17,23-*tetrakis*(2,5-Dimethoxy-4-methylphenyl)-25,26,27,28-tetrapropoxycalix[4]arene (1,3-Alternate Conformer) (E)**. Yield: (75%); mp 388–390 °C (CH_2_Cl_2/_MeOH). ^1^H NMR (CDCl_3_) δ: 0.77 (t, *J* = 7.41 Hz, 12H), 1.65 (sextet, *J* = 7.6 Hz, 8H), 2.29 (s, 12H), 3.38 (s, 12H), 3.62 (t, *J* = 7.44 Hz, 8H), 3.69 (s, 12H), 3.76 (s, 8H), 6.56 (s, 4H), 6.77 (s, 4H), 7.17 (s, 8H); ^13^C NMR (CDCl_3_), δ: 11.05, 17.07, 24.50, 37.99, 55.41, 57.12, 74.02, 112.29, 115.03, 125.21, 128.73, 130.65, 130.93, 132.51, 149.34, 151.20, 155.46. MS (ESI) calculated mass for the parent C_76_H_88_O_12_ 1193.54, found 1193.57 [M + 1]^+^. Analysis for C_76_H_88_O_12_ (1193.5) Calcd. %C, 76.48; H, 7.43; Found %C, 77.87; H, 6.92.

### 3.4. X-ray Crystallography

Accession Codes: CCDC 2194839–2194843 contain the supplementary crystallographic data for this paper. These data can be obtained free of charge via www.ccdc.cam.ac.uk/data_request/cif, or by emailing data_request@ccdc.cam.ac.uk, or by contacting The Cambridge Crystallographic Data Centre, 12 Union Road, Cambridge CB2 1EZ, UK; fax: +44-1223-336033. (Accessed on 4 August 2022).

## 4. Conclusions

We have described herein the preparation of various calixarene derivatives that incorporated with 2,5-dimethoxytolyl as an electron donor. This electro-active aryl groups in calixarene ethers allow the hole to be delocalized over the aryl groups of the calixarene core to yield an electron-deficient cavity. Accordingly, the modified calixarene derivatives allow preparation of a stable mono- and tetra-cation radical salt through a chemical or an equivalent electrochemical oxidation. The cation radicals of calixarenes are stable indefinitely at ambient temperatures and can be readily characterized by UV-vis-NIR spectroscopy. The isolation of a stable cation radical binds a single molecule of nitric oxide deep within its cavity with remarkable efficiency. A careful delineation of these two modes of charge delocalization in various calixarene cation radicals will be further investigated by a careful Marcus-Hush treatment of the NIR absorption bands.

## Data Availability

Not applicable.

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
