# Peer review of "Design, Synthesis, Electronic Properties, and X-ray Structural Characterization of Various Modified Electron-Rich Calixarene Derivatives and Their Conversion to Stable Cation Radical Salts"

_molecules, 2022, doi:10.3390/molecules27185994_

Round 1

Reviewer 1 Report

The authors report the syntheses, electrochemical properties, and X-ray structures of some electron-rich calix[4]arene ether derivatives, and the preparations and spectroscopic characterizations (UV-vis-NIR) of their oxidized species.  And also they studied the reactions of their oxidized species with NO radical.  It is interesting but has many problems.

The authors have deposited the crystallographic data of the analyzed calix[4]arene ether derivatives at CCDC, but it is necessary to describe them in the Supporting Information for the convenience of the reader.  In particular, experimental details of crystallographic data for all crystals analyzed and bonding parameters for all structures should be presented as SI tables.  Also, please provide atomic labels on the crystal structure diagram.

Elemental analyses of the synthesized calix[4]arene molecules should be reported in the manuscript.  The syntheses of the important calix[4]arene are reported and their characterizations are done by 1H-NMR, 13C-NMR, and ESI-MS.  The spectra are clear, so there is no doubt about them, but elemental analysis is still the minimum necessary and important data in chemical synthesis.

The authors should assign peaks of 1H and 13C NMR spectra in SI even if they are tentative.

In the CV data, only oxidation waves are tabulated and discussed, but I would like to see the reduction waves tabulated as well.

The reaction between oxidizing species and NO radicals should be well described and discussed. The Abstract describes the reaction with NO, but it is not described at the Conclusion section.  The X-ray structure of the NO adduct is also obtained, but it is not described or discussed.

There are several errors of sentences in the manuscript. The manuscript should be written with careful attention.

Author Response

1) The Data of X-rays were fixed according to the reviewer notes.

2) The elemental analysis data were recorded for the essential compounds

3) The NMR spectra were assigned for the essential compounds

4) Electrochemical oxidation potentials were necessary for this study

5) The reaction between oxidizing species and NO radicals was well described and discussed (It is worth mentioning that, this work is an extension of our previous work, https://doi.org/10.1021/ja0454900 & https://doi.org/10.1002/1521-3773(20000616)39:12<2123::AID-ANIE2123>3.0.CO;2-4 that is why the X-ray structure of the NO adduct is not described or discussed but we use it as an example of proof of concept.

6) The manuscript was reviewed carefully, and errors were fixed.

Reviewer 2 Report

Dear Authors

Good arraenged with good novelity paper containing complete characterizations!

Author Response

Thank you 

Reviewer 3 Report

Dear Author

The following minor revisions are recommended for improving the manuscript:

1-In page 3, scheme 2, reaction conditions, second c and d should be changed to e and g, respectively.

2-In page 3, scheme 3, reaction conditions, second b and c and d should be changed to c, d and f, respectively.

3-In page 10, line 237, first sentence "Binding of Nitric Oxide (NO) to Various Modified Calixarene Donors" should be bold, and the second "Binding of Nitric Oxide (NO) to Various Modified Calixarene Donors" is repetitive and should be omitted.

4-In page 11, the (MA+.) and (NAP+.) cation radicals are not in the same writing styles. Choose one of them and apply it for whole manuscript. 

5-In page 12, line 315, change UVvis to UV-Vis.

6-In page 13, line 391, change momo to mono.

7-In page 13, line 414, add general procedure for di-bromination of calix[4]arene derivatives.

8-In page 15, line 500, change "preparation various" to "preparation of various".

9-In page 16, line 528, correct the DOI.

10- Generally, the referencing style is not clear. For example reference 9 and 10 includes seven individual papers! which is very confusing. Change the citation manner and divide the combined references into separate ones.

Author Response

1-In page 3, scheme 2, reaction conditions, second c and d should be changed to e and g, respectively.

fixed

2-In page 3, scheme 3, reaction conditions, second b and c and d should be changed to c, d and f, respectively.

fixed

3-In page 10, line 237, first sentence "Binding of Nitric Oxide (NO) to Various Modified Calixarene Donors" should be bold, and the second "Binding of Nitric Oxide (NO) to Various Modified Calixarene Donors" is repetitive and should be omitted.

fixed

4-In page 11, the (MA+.) and (NAP+.) cation radicals are not in the same writing styles. Choose one of them and apply it for whole manuscript. 

fixed

5-In page 12, line 315, change UVvis to UV-Vis.

fixed

6-In page 13, line 391, change momo to mono.

fixed

7-In page 13, line 414, add general procedure for di-bromination of calix[4]arene derivatives.

fixed

8-In page 15, line 500, change "preparation various" to "preparation of various".

fixed

9-In page 16, line 528, correct the DOI.

fixed

10- Generally, the referencing style is not clear. For example reference 9 and 10 includes seven individual papers! which is very confusing. Change the citation manner and divide the combined references into separate ones.

fixed